# A Collaborative Medication Review Including Deprescribing for Older Patients in an Emergency Department: A Longitudinal Feasibility Study

**DOI:** 10.3390/jcm9020348

**Published:** 2020-01-27

**Authors:** Morten Baltzer Houlind, Aino Leegaard Andersen, Charlotte Treldal, Lillian Mørch Jørgensen, Pia Nimann Kannegaard, Luana Sandoval Castillo, Line Due Christensen, Juliette Tavenier, Line Jee Hartmann Rasmussen, Mikkel. Zöllner Ankarfeldt, Ove Andersen, Janne Petersen

**Affiliations:** 1Clinical Research Centre, Copenhagen University Hospital Amager and Hvidovre, 2650 Hvidovre, Denmark; 2The Capital Region Pharmacy, 2730 Herlev, Denmark; 3Department of Drug Design and Pharmacology, University of Copenhagen, 2100 Copenhagen, Denmark; 4Department of Geriatric Medicine, Copenhagen University Hospital Herlev and Gentofte, 2900 Hellerup, Denmark; 5Department of Geriatrics, Copenhagen University Hospital Bispebjerg and Frederiksberg, 2400 Copenhagen, Denmark; 6Research Unit for General Practice, 8000 Aarhus, Denmark; 7Department of Psychology and Neuroscience, Duke University, Durham, NC 27708, USA; 8Copenhagen Phase IV unit (Phase4CPH), Center of Clinical Research and Prevention and Department of Clinical Pharmacology, Copenhagen University Hospital Bispebjerg and Frederiksberg, 2000 Copenhagen, Denmark; 9Emergency Department, Copenhagen University Hospital Amager and Hvidovre, 2650 Hvidovre, Denmark; 10Section of Biostatistics, Department of Public Health, University of Copenhagen, 1014 Copenhagen, Denmark

**Keywords:** medication review, deprescribing, Medication Appropriateness Index, potentially inappropriate medication, polypharmacy, clinical pharmacy, geriatric, emergency department

## Abstract

Medication review for older patients with polypharmacy in the emergency department (ED) is crucial to prevent inappropriate prescribing. Our objective was to assess the feasibility of a collaborative medication review in older medical patients (≥65 years) using polypharmacy (≥5 long-term medications). A pharmacist performed the medication review using the tools: Screening Tool of Older Persons’ potentially inappropriate Prescriptions (STOPP) criteria, a drug–drug interaction database (SFINX), and Renbase^®^ (renal dosing database). A geriatrician received the medication review and decided which recommendations should be implemented. The outcomes were: differences in Medication Appropriateness Index (MAI) and Assessment of Underutilization Index (AOU) scores between admission and 30 days after discharge and the percentage of patients for which the intervention was completed before discharge. Sixty patients were included from the ED, the intervention was completed before discharge for 50 patients (83%), and 39 (61.5% male; median age 80 years) completed the follow-up 30 days after discharge. The median MAI score decreased from 14 (IQR 8-20) at admission to 8 (IQR 2-13) 30 days after discharge (*p* < 0.001). The number of patients with an AOU score ≥1 was reduced from 36% to 10% (*p* < 0.001). Thirty days after discharge, 83% of the changes were sustained and for 28 patients (72%), 1≥ medication had been deprescribed. In conclusion, a collaborative medication review and deprescribing intervention is feasible to perform in the ED.

## 1. Introduction

Medication optimization in older patients is complex and challenging [1]. Previous observational studies indicate that adverse drug reactions due to overprescribing, under-prescribing, or potentially inappropriate medications (PIMs) are responsible for 1.5%–15% of emergency department (ED) visits and unplanned hospitalizations for older patients [2]. Older medical patients admitted to the ED are at particularly high risk for adverse drug reactions (ADRs), as up to 80% are exposed to polypharmacy (≥5 prescribed medications) and 85% receive at least one PIM [3]. Medication prescribing to frail older patients is complex due to variations in aging-related physiological changes, comorbidities, and altered pharmacokinetic and pharmacodynamic responses to medication [4]. A systematic review of ADRs found that 45% of all adverse drug reactions leading to hospitalization are preventable [5]. ADRs can result in increased hospitalization, costs, morbidity and, in some cases, death [5,6,7,8]. Pharmacist-led medication reviews have been proposed as an important part of the solution to the medication-related problems [9].

Some older patients are prescribed medications throughout their lives, but the appropriateness of continuing each medication is rarely evaluated in relation to the patient’s changing health condition [1,10,11,12]. Therefore, dose reduction and discontinuation of inappropriate medication (deprescribing) is an important focus of medication reviews in older patients [13]. Since more than half of all patients admitted to the ED in Denmark are discharged without being transferred to another department [14], it is very relevant to perform the medication reviews in the ED. Many medication review tools have been developed to identify PIMs in older patients, such as the validated Screening Tool of Older Persons’ potentially inappropriate Prescriptions (STOPP) criteria [15]. However, there is contradictory evidence regarding the efficacy of medication reviews for older patients in the ED [16,17,18,19,20]. While studies with collaboration between pharmacists and physicians in multidisciplinary teams on wards have been shown to significantly decrease prescribing inappropriateness in older patients across sectors [21,22,23], this approach has never been tested in an ED.

The aim of the study was therefore to determine the feasibility of a collaborative medication review and deprescribing intervention by a pharmacist and geriatrician for older patients hospitalized in the ED. The primary objective was to evaluate the feasibility of the intervention reflected by the change in the Medication Appropriateness Index (MAI) score from hospital admission to 30 days after discharge. Secondary objectives were to evaluate the change in underutilization (the omission of appropriate medication) using the Assessment of Underutilization Index (AOU) score from hospital admission to 30 days after discharge, the percentage of patients where it was possible to complete the intervention before discharge without affecting hospitalization time, and the agreement between the pharmacist’s suggestions and the geriatrician’s intervention.

## 2. Experimental Section

### 2.1. Ethics Approval and Trial Registration

This study is part of a set of studies named FAM-CPH that investigate medication, malnutrition, biological aging, and inflammation among healthy individuals and acutely ill older patients admitted to the ED. FAM-CPH is registered at Clinical Trials.gov (identifier: NCT03052192). The present study was conducted in accordance with the Declaration of Helsinki, and all participants gave written informed consent. The study was approved by the Danish Data Protection Agency (AHH-2016-067) and the Research Ethics Committees for the Capital Region of Denmark (H-16038786).

### 2.2. Setting

The primarily tax-funded Danish health care system is based on the principles of free and equal access to health care and grants universal health coverage to all citizens. Copenhagen University Hospital, Hvidovre, Denmark, covers 10 municipalities with approximately 550,000 citizens. The hospital has approximately 14,000 medical admissions each year—of which, 85% are acute admissions. The ED at Hvidovre Hospital has a 29 bed medical ward handling all acute admissions, and a separate emergency room (ER) handling all minor injuries and traumas. Patients are referred to the ED by general practitioners (GPs), medical helpline or emergency call. The patients can be hospitalized for up to three days before discharge or transferred to a specialized medical ward. In daily practice, the ED has permanently affiliated clinical pharmacists and geriatricians.

### 2.3. Design and Patients

This was a longitudinal feasibility study with older medical patients in the ED. Patients were enrolled at time of ED admission from November 2016 to August 2017 with a follow-up visit at the patients’ home 30 days after discharge. The inclusion criteria were: (i) admitted to the medical ED, and (ii) age ≥65 years. The exclusion criteria were: (i) inability to understand Danish, (ii) inability to cooperate physically or cognitively, (iii) in isolation, (iv) not Caucasian, (v) admitted due to suicide attempt or terminally ill, and (vi) less than five long-term medications. Patients were screened each morning (only on weekdays) from a randomly ordered computer-generated list that included all patients meeting the inclusion criteria.

### 2.4. Baseline Data Collection within 24 h after Admission

Demographic information was recorded at inclusion. Furthermore, physical parameters including weight, height, and handgrip strength assessed in the dominant hand using a handheld dynamometer (Saehan, Digi-II) [24] were recorded. Health-Related Quality of Life (HRQOL) was assessed with the 5 level EuroQol 5 dimensional questionnaire (EQ-5D-5L) [25]. Cognitive status was assessed with the Short Orientation-Memory-Concentration (OMC) test [26]. Nutritional status was assessed with the Mini Nutritional Assessment-Short Form (MNA) consisting of 6 questions providing a score from 0 to 14 (<11 is considered malnourished for the purpose of this study) [27]. Cumulative organ dysfunction was assessed by a frailty index (FI-OutRef), calculated as number of laboratory results outside of reference interval for 17 standard biomarkers routinely analyzed at admission [28]: C-reactive protein (CRP), white blood cell count, neutrophils, hemoglobin, mean corpuscular hemoglobin concentration, mean corpuscular volume, thrombocytes, creatinine, blood urea nitrogen, sodium, potassium, albumin, alanine aminotransferase, alkaline phosphatase, lactate dehydrogenase, bilirubin, and coagulation factors II, VII, and X. All 17 biomarkers were routinely analyzed at the Department of Clinical Biochemistry. Estimated glomerular filtration rate (eGFR) was calculated using equations from Chronic Kidney Disease Epidemiology (CKD-EPI) based on creatinine [29].

### 2.5. Intervention

The intervention consisted of an initial medication reconciliation upon admission, a collaborative medication review intervention focused on deprescribing, and a final ready-to-discharge medication reconciliation. All steps were completed within the first 24 h of hospitalization by a senior clinical pharmacist and a senior geriatrician.

#### 2.5.1. Medication Reconciliation upon Admission

For patients included in the study, the clinical pharmacist obtained the most complete medication list possible, starting with the initial medication list that was used upon admission. An initial medication list was composed by the admitting physician in the ED using the Shared Medication Card (SMC), a central database of all Danish citizens containing information about all medications prescribed and dispensed during the previous two years [30], as well as a patient interview at admission. The initial list was checked by a clinical pharmacist with reference to the SMC and the hospital’s electronic patient record. Next, a structured interview with the patient was performed to confirm use of all prescribed or over-the counter medications. In cases where a patient could not provide reliable information, the medication list was confirmed with a caregiver of the patient, such as a relative, GP, home nurse, nursing home staff, or community pharmacy. The final medication list was then documented in the electronic patient record by the clinical pharmacist. Any discrepancies between the initial and final medication list was noted and described to the providing geriatrician, who then updated the patient’s electronic prescription accordingly. Discrepancies were classified as (a) omission on initial list, (b) discontinuation, (c) incorrect dose, or (d) incorrect dosing interval with clinical impact.

#### 2.5.2. Medication Review and Deprescribing Intervention

After the final medication list was obtained and laboratory data became available, a structured, patient-oriented medication review was conducted by a clinical pharmacist. Each medication was reviewed by a clinical pharmacist for (1) agreement with current national guidelines regarding choice of medication, dose, and dosing interval; (2) whether the goal of treatment had been met; and (3) inadequate treatment of any current diagnoses/condition. The STOPP criteria version 2 was used to identify PIMs [15]. Prescribing recommendations based on eGFR were obtained from Renbase^®^ [31] and a national database (“pro.medicin.dk”), which contains information on medication and treatment guidelines [32]. Moreover, medications were labeled as “renal risk” if dose adjustments were recommended at eGFR ≤ 90 mL/min/1.73 m^2^. Drug interactions were checked with the SFINX-database [33] and a national interaction database (“interaktionsdatabasen.dk”) [34].

Finally, each prescribed medication was assessed for appropriateness based on indication for treatment, dose (considering age, kidney and liver function, etc.), adverse drug reactions, therapeutic duplication, dosing interval, formulation and strength, drug interactions, contraindications, precautions, and specific patient characteristics. If medications were deemed unnecessary, the treatment was proposed to be discontinued. The clinical pharmacists were not authorized to implement changes in the patients’ medication after having performed the medication review but documented the proposed recommendations in the electronic patient record for the geriatricians to review. Any recommendations by the clinical pharmacist were afterward discussed with the geriatrician in the ED, who decided whether to accept, reject, or alter the recommendation. All medication changes were ultimately documented by the geriatrician in the electronic patient record and either (i) implemented in the ED, (ii) accepted but not implemented in the ED, (iii) communicated to a specialized hospital ward and/or the patient’s GP, or (iv) rejected.

#### 2.5.3. Ready-to-Discharge Medication Reconciliation

Immediately following medication review of each patient, a ready-to-discharge medication reconciliation including updated medication list and SMC was prepared by the geriatrician as a part of the intervention. Upon hospital discharge, the treating physician then determined whether to accept this medication reconciliation or propose further changes and provided each patient with a copy of their updated medication list.

### 2.6. Data Collection 30 Days after Discharge

A follow-up visit 30 days after discharge was conducted in the patient’s home by a research assistant trained in collection of a full medication list and measuring HRQOL. The research assistant performed a structured interview with the patient to confirm the current use of all prescribed or over-the counter medications. Again, in cases where a patient could not provide reliable information, the medication list was confirmed by a caregiver, such as a relative or nursing home staff. In case of any doubt related to the medication list these were clarified by a clinical pharmacist over the phone after the follow-up visit.

### 2.7. Outcome Measures

The primary outcome was the change in the MAI from hospital admission to 30 days after discharge. Secondary outcomes were AOU scores, the percentage of patients where it was possible to complete the intervention before discharge without affecting hospitalization time and frequency of the geriatrician agreement with the pharmacist’s proposed recommendations and frequency of the geriatrician implementing of the pharmacist’s proposed recommendations.

#### 2.7.1. The Medication Appropriateness Index and the Assessment of Underutilization Index

The MAI consists of 10 criteria addressing different aspects of each prescription, including indication, effectiveness, dose, direction, practical direction, drug–drug interaction, drug–disease interaction, duplication, duration of therapy, and cost [35]. Each criterion has operational definitions that instruct the evaluator to rate a medication as (A) appropriate, (B) marginally appropriate, (C) inappropriate, or (Z) do not know. For this study, each medication was labeled as appropriate (rating A or B), inappropriate (rating C), or unknown (rating Z) [36]. Each of the criteria received a score between 0 and 3 according to standard protocol. Each prescription could have a potential score between 0 and 18 [37], where 0 represents appropriate prescribing and higher scores indicate a greater degree of inappropriateness. The total MAI score was then calculated as the sum of individual MAI scores for each medication used by the individual patient [38] at three time points: admission, after intervention, and 30 days after discharge. A change of five points in the MAI score was considered as clinically relevant [21]. The MAI has been found to be a reliable, valid, standardized instrument for the evaluation of multiple elements of drug therapy prescribing for a variety of medications, clinical conditions, and settings [35,39,40,41].

The AOU identifies omitted medication prescribing despite being indicated with potential benefit [42]. The evaluator requires a full medication list as well as a list of established medical conditions to apply one of three ratings for each condition: (A) no omission, (B) marginal omission, or (C) omission of an indicated medication without contraindication. For this study, each condition was labeled as no omission (rating A or B) or omission of an indicated drug (rating C) [36]. The number of untreated conditions (AOU score) was rated for each patient at three time points: admission, after intervention, and 30 days after discharge. The outcome measure was the proportion of patients with an AOU score ≥1 at the given time points.

#### 2.7.2. Evaluation of the MAI and the AOU

For the first 15 included patients (165 medications at admission), MAI and AOU scores were independently evaluated by a senior clinical pharmacist (>5 years of clinical experience), senior geriatrician (>5 years of clinical experience), and a junior geriatrician (<5 years of clinical experience). The evaluators only evaluated the outcomes and were not involved in the study’s intervention. The evaluators were trained in using the MAI and the AOU and blinded to both proposed and accepted medication changes during the intervention. The kappa coefficients (κ) between the three evaluators were ≥0.82 for MAI criteria and ≥0.93 for AOU criteria which is considered as almost perfect agreement [43]. Discrepancies in MAI and AOU scores between the evaluators were reviewed in person and consensus was reached. For the remaining 24 included patients (289 medications at admission) (Figure 1), MAI and AOU scores were independently evaluated by the senior pharmacist and the senior geriatrician using a similar procedure. There was also close agreement between the two evaluators for these scores, with κ-coefficients of ≥0.85 for MAI criteria and ≥0.90 for AOU criteria.

#### 2.7.3. Feasibility to Complete the Intervention before Discharge

The collaborative intervention was considered feasible if the pharmacists’ and geriatricians’ work did not affect hospitalization time. If a patient was informed of discharge before the intervention was complete, then the intervention was discontinued. Feasibility of the intervention was evaluated based on the percentage of patients who completed the intervention without affecting hospitalization time: very low (<25%), low (>25% to< 50%), moderate (>50% to <75%), or good (>75%) feasibility [24].

### 2.8. Sample Size Calculation

A 5 point decrease in the MAI score between admission and 30 days after discharge was considered to be a clinically relevant improvement in medication prescribing [21]. It was determined that a sample size of 40 participants, including an expected dropout rate of 15% at 30 days after discharge, was necessary to detect a difference in the primary outcome with 80% power using a one-sample t-test (two-tailed α = 0.05) and variation set conservatively to 10.0 [21,23,44,45].

### 2.9. Statistical Analysis

All patient characteristics are presented as the median and interquartile range (IQR), since not all parameters were normally distributed. The primary outcome of the study was the change in the MAI score between admission and 30 days after discharge. MAI scores are presented as the median and IQR. Change in MAI scores at admission, after intervention, and 30 days after discharge were analyzed with the Wilcoxon signed rank-sum test. The two-proportions z-test was used for statistical comparisons of the number of patients with an AOU score ≥1 at admission, after intervention, and 30 days after discharge. Likewise, z-test test was used to compare the number of patients with ≥1 renal risk medication dosed higher than recommended at the different time points. Patients’ total numbers of medications and HRQQL are presented as means with standard deviation (SD). Differences in patients’ total numbers of medication and HRQOL between time points was analyzed with a paired t-test. For all statistical tests, *p* < 0.05 was considered statistically significant. All calculations and statistical analyses were performed in SAS Enterprise Guide 7.1. (SAS Institute Inc., Cary, NC, USA). Figures were created in RStudio 3.2.3., (Integrated Development for R. RStudio, Inc., Boston, MA, USA.).

## 3. Results

A total of 550 older patients were screened when admitted to the ED. Of these, 337 patients were excluded (Figure 1). Inability to cooperate cognitively was the primary reason for exclusion (56%). Of the remaining 212 eligible patients, all were asked to participate, and 128 consented to participation. Ninety-one (71%) of the 128 patients used ≥5 long-term medications daily (Figure 1). Of the 91 patients, 71 were eligible for the intervention. A further 11 patients were excluded due to lack of pharmacist or geriatrician to perform the intervention. Ten patients were excluded due to discharge before the intervention was finalized. Thereby, the intervention was feasible for 50 out of 60 patients (83%) without increasing hospitalization time. A total of 39 patients completed both the intervention and follow-up 30 days after discharge.

Patient characteristics for the final study population (*n =* 39) are shown in Table 1. Among included patients, 62% were male and the median age was 80 years. Median length of hospitalization was 3 days, and approximately 40% of patients were discharged directly from the ED. All patients used at least one medication listed in the STOPP version 2 criteria, and 82% of patients used ≥10 medications at baseline.

### 3.1. Medication Reconciliation upon Admission

In total, 77 discrepancies distributed over 29 patients (74%) were identified by the clinical pharmacist between the initial and final medication list, with an average of 2.0 (77/39) discrepancies per patient. Of these, the geriatrician agreed with 77 (100%) of the discrepancies and recorded 75 (98%) of the identified discrepancies in the electronic medical record. The geriatrician considered that the remaining two discrepancies should not be implemented due to the patient’s current condition. Of the 75 implemented discrepancies, 26 (35%) were due to incorrect dose, 24 (32%) were due to omission on initial list, 18 (24%) were due to discontinuation, and 7 (9%) were due to incorrect dosing interval with clinical impact.

### 3.2. Medication Review and Deprescribing Intervention

In total, 131 prescription changes were recommended by the clinical pharmacist at hospital admission. Of these, the geriatrician agreed with 106 (81%) of the changes and implemented 80 (61%). The geriatrician implemented an additional 15 changes, resulting in 95 implemented changes. An average of 2.4 prescription changes per patient, including at least one change in 31 patients (79%), was implemented by the geriatrician. Thirty days after discharge, 79 (83%) of these changes were sustained.

#### 3.2.1. The Medication Appropriateness Index and the Assessment of Underutilization Index

The appropriateness of prescribed medication was assessed by the total MAI score at each time point for each patient and are shown in Figure 2. The median MAI score decreased from 14 (IQR 8–20) at admission to 6 (IQR 4–10) after intervention and 8 (IQR 2–13) 30 days after discharge. Median MAI scores after the intervention and 30 days after discharge were both significantly lower than the median MAI score at admission (*p <* 0.001). There was no significant difference in the MAI score after the intervention and 30 days after discharge (*p =* 0.76). Overall, the results from Figure 2 show that the intervention increases the prescribing appropriateness and the changes were sustained 30 days after discharge. Change in the MAI score across time points are shown in Table 2. Between admission and 30 days after discharge, 25 patients (64%) had a reduction in the MAI score ≥5 points considered as a clinically relevant change. Furthermore, 16 patients (41%) had a reduction >10 points, while 4 patients (10%) had increased MAI score. Specific MAI criteria deemed inappropriate at each time point are shown in Table 3. At admission, 14% (*n =* 62) of medications had been prescribed at incorrect doses, 11% (*n =* 48) were not indicated, and 9% (*n =* 41) were not effective. Thirty days after discharge, this decreased to 7% (*n =* 33), 7% (*n =* 31), and 7% (*n =* 31) of medications, respectively. Out of all prescribed renal risk medications (*n =* 149), 12% (*n =* 18) were prescribed at higher doses than recommended before the intervention. In total, 36% (*n =* 14) of patients received at least one renal risk medication with a dosing discrepancy at admission, while these numbers were reduced to 5% (*n =* 2) 30 days after discharge (*p <* 0.001). Proton pump inhibitors; nonbenzodiazepines (z-drugs), benzodiazepines, first-generation antihistamines; antihypertensives; opioids (including Tramadol and Codeine); and nonsteroidal anti-inflammatory drugs (NSAIDs) were the respective drug classes where the intervention showed the biggest improvement in the MAI score between admission and 30 days after discharge.

Changes in AOU scores between admission and 30 days after discharge are also shown in Table 2. In total, 14 patients (36%) had at least 1 untreated condition at admission. This decreased to three patients (8%) after intervention and four patients (10%) 30 days after discharge. Significantly fewer patients had untreated conditions after intervention and 30 days after discharge compared to admission (*p <* 0.001), while there was no difference between after intervention and 30 days after discharge (*p =* 0.97). In total, the intervention resulted in prescription of 12 new medications to nine patients. At admission, the most frequently under-prescribed clinically indicated medication were calcium supplements (*n =* 4), laxatives (*n =* 3), and potassium supplements (*n =* 2).

#### 3.2.2. Deprescribed Medication, Number of Medications and Quality of Life

In total, the intervention deprescribed 66 medications in 33 patients (84%) (Table 4). Of these, 36 medications were discontinued, and 30 medications were dose-reduced. Fifteen patients had one deprescription (38%), nine patients (23%) had two deprescriptions, and nine patients (23%) had ≥3 deprescriptions. Thirty days after discharge, 62 of 66 deprescriptions (94%) were sustained. Overall, this means that we—in average—deprescribed 1.6 medications per patients. The average (SD) number of medications at admission was 11.9 (3.2), after intervention 11.3 (3.0), and at 30 days follow-up 11.5 (3.3). The difference in total number of medications was significantly lower between admission and after intervention (*p =* 0.039). Thirty days after discharge there was no difference in the total number of medications compared to admission (*p =* 0.36). Finally, no significant difference was found between the mean (SD) HRQOL 0.64 (0.23) at admission and 0.71 (0.18) at 30 days after discharge (*p =* 0.42) (*n =* 38).

## 4. Discussion

### 4.1. Main Findings and Implications

In the current study, we tested the feasibility of a collaborative medication review intervention between pharmacists and geriatricians, as well as its impact on prescribing appropriateness in older patients with polypharmacy admitted through the ED. There were two primary findings which indicate feasibility. First, the intervention was associated with significantly lower MAI and AOU scores 30 days after discharge compared to admission. Second, the intervention was feasible to complete for 83% of patients exposed to the intervention without affecting duration of hospitalization.

Overall, the appropriateness of prescribing improved significantly following medication review. Compared to admission, the MAI score decreased by a median of 8 points immediately after intervention and six points at 30 day follow-up. In addition, approximately 40% of patients had a decrease in the MAI of 10 points or more between admission and 30 days after discharge. On average, we deprescribed 1.6 medications per patient without finding any difference in HRQOL. Because of new treatments initiated during hospitalization, we observed no difference in total number of medications at the 30 day follow-up compared to admission. Many older patients with polypharmacy are admitted to the ED, and approximately 40% of the patients in our study were discharged directly from the ED. Despite this rapid patient turnover, our simple recruitment process demonstrates practical feasibility of the intervention as 83% of all our prescribing changes being sustained 30 days after discharge.

### 4.2. Results in Context of Other Studies

To our knowledge, this is the first study to demonstrate feasibility of a collaborative medication review and deprescribing intervention in an ED. We found on average two discrepancies per patients between the initial medication list from admission and the final list recorded by the clinical pharmacist. These results are consistent with existing literature [46,47] and emphasize the necessity of medication reconciliation as a part of a medication review intervention. The geriatrician agreed with 81% and implemented 61% of changes recommended by the clinical pharmacist, which is within the implementation range of 40%–90% reported by other studies [48,49,50]. Considering the acute condition of patients and rapid patient turnover in the ED, our reported implementation rate of 61% indicates successful cross-functional collaboration between pharmacists and geriatricians. Increasing this implementation rate would require a more complex intervention with multiple intervention times, closer cooperation with GPs and more time. A study by Larsen et al. found that only 64% of all prescribing changes in Danish hospitalized geriatric patients were accepted by GPs [51]. In a separate study, the same authors concluded that this low acceptance rate was primarily due to miscommunication between the hospital and primary care physicians [52]. In our study, 83% of prescribing changes were sustained after discharge, which is likely the result of ready-to-discharge medication reconciliation, patient involvement and written communication to GPs.

Our MAI score results are comparable to those reported by randomized controlled trials using a collaborative approach to reduce inappropriate prescribing in older patients. In a Swedish intervention study performed in an internal medicine ward, the MAI score decreased from 11.4 at admission to 6.4 at discharge [21]. In another Swedish intervention study investigating the effect of the Lund Integrated Medicines Management model in internal medicine wards, MAI scores decreased from 12.5 to 3.5 [23]. Moreover, in a Belgian intervention study involving geriatric evaluation and pharmaceutical management, MAI scores decreased from 24.1 to 7.1 [22]. In a South Australian intervention study, Crotty et al. reported that the mean MAI score decreased from 7.4 at admission to 3.5 after discharge to a long-term care facility [53]. Finally, Gallagher et al. found that a physician-driven medication review based on the STOPP and Screening Tool to Alert to Right Treatment (START) criteria in an Irish ED resulted in a MAI score reduction from 10 to 3 after intervention [36]. None of these studies found significant differences in the MAI score for control groups who did not receive the intervention [21,22,23,36,53]. Differences in MAI score improvement between these studies can be attributed to the medical setting, number and types of medication, degree of collaboration between physicians and pharmacists, and clinical judgment of the raters [39]. Our reported decrease of 6 MAI points from admission to 30 days after discharge is comparable to these results from international, randomized, controlled trials. Discrepancies between the success of our intervention and those using a collaborative approach [21,22,23,53] can be explained by two main differences: i) our intervention was performed in an emergency setting with rapid patient turnover whereas the other studies were performed in inpatient wards, and ii) our intervention included an integrated ready-to-discharge medication reconciliation procedure whereas the others had an independent medication reconciliation intervention at discharge.

The majority of the pharmacist’s recommendations during medication review that resulted in MAI score improvement was related to deprescribing. This is consistent with findings from previous pharmacist-led medication review studies [22,53]. Dosing recommendations were typically due to age-related changes in drug clearance, such as reduced renal function. More than one-third of patients in our study had at least one medication prescribed at a dose higher than recommended according to their renal function at the time of inclusion. This finding is notably higher than in other Danish studies focusing on dosing of renal risk medication in hospital [54,55]. A Swedish study concluded that one third of all adverse drug reactions leading to hospitalization is related to improper dose adjustment according to kidney function [56], which highlights the importance of careful dosing for renal risk medications. Approximately two-thirds of patients in our study were malnourished, and it is well known that creatinine-based GFR estimates can be inaccurate in malnourished patients since creatinine production is highly dependent on muscle mass [57]. In these patients, a medication review may be particularly relevant and use of alternative eGFR biomarkers such as cystatin C may be more accurate and lead to fewer dosing errors [58,59].

### 4.3. Strengths and Limitations

The main strength of this study is the integration into daily context-specific conditions and use of blinded MAI and AOU score evaluators who had almost perfect agreement between ratings. This study also has several limitations. First, the MAI tool was developed to measure medication appropriateness, not clinical outcomes. However, a higher MAI score has been shown to be significantly associated with poor clinical outcomes such as unplanned hospitalization, adverse drug events, drug-related hospitalization and reduced HRQOL [23,60,61,62]. Second, since this is a feasibility study, it was designed at a single center without a control group. Inclusion of controls would provide information about how MAI scores are affected by standard care in the ED. Third, we have only one follow-up time point at 30 days after discharge and therefore do not know how long the effect of the intervention is sustained. One could imagine that 30 days is a little too short for all patients to have been seen by their GP. Finally, our study excluded patients with cognitive impairment, though this group of patients is frequently admitted to the ED and may also benefit from medical review. To address these limitations, future studies should be performed in more diverse patient groups and over longer time periods. For that reason, we have designed and started the randomized controlled trial concerning Optimization of Nutrition and Medication for Acutely Admitted Older Medical Patients (OptiNAM study) registered at Clinical Trials.gov (identifier: NCT03741283).

## 5. Conclusions

Among older patients with polypharmacy in the ED, a collaborative medication review and deprescribing intervention resulted in significant and sustained improvements in both MAI and AOU scores between hospital admission and 30 days after discharge. On average, 1.6 medications were deprescribed per patient during the intervention, while the total number of medications per patient remained unchanged. Despite rapid patient turnover, we observed successful cross-functional collaboration between pharmacists and geriatricians as well as an overall good feasibility for our intervention. Future studies should investigate the effect of such interventions in randomized controlled trials to identify which subgroups of patients benefit most.

## Figures and Tables

**Figure 1 jcm-09-00348-f001:**
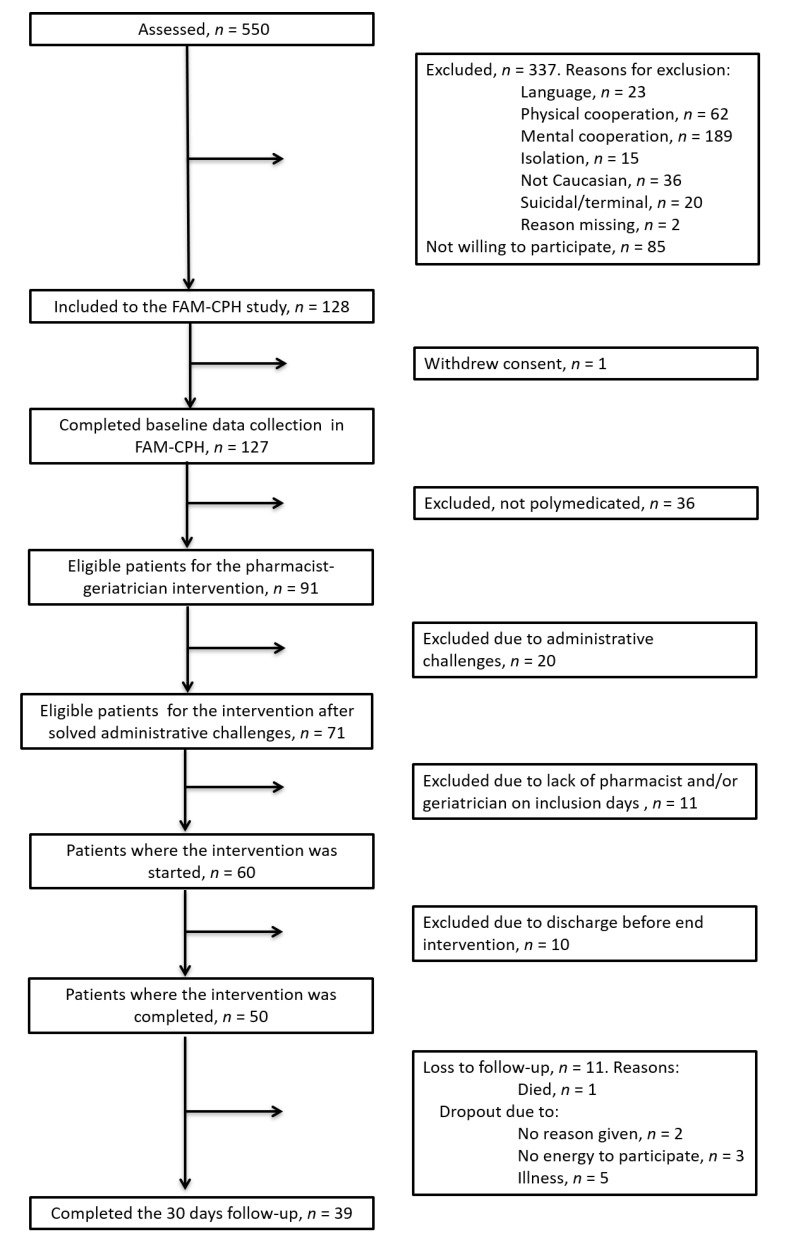
Flowchart of inclusion of patients in the study. In total, 39 patients completed the 30 days follow-up.

**Figure 2 jcm-09-00348-f002:**
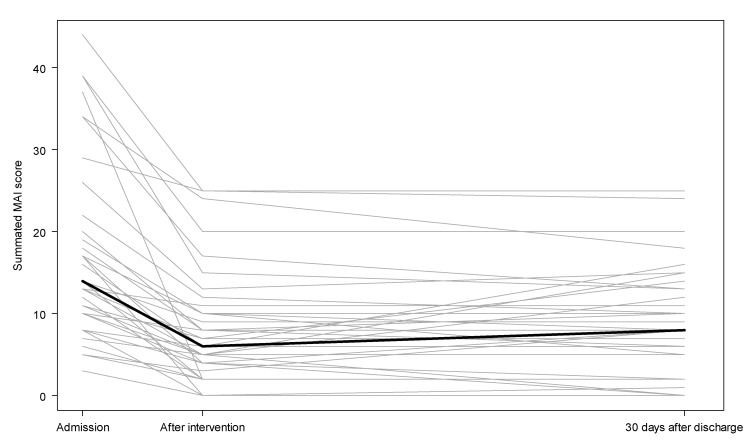
Summated Medication Appropriateness Index (MAI) scores at admission, after intervention, and at 30 days after discharge (*n =* 39). Light grey lines represent each patient, solid black line is the median.

**Table 1 jcm-09-00348-t001:** Characteristics of the included older medical patients (*n =* 39).

Demographic Data	*n*	Median (IQR) or *n* (%)
Male sex, *n* (%)	39	24 (61.5)
Age (years)	39	80 (73—87)
Body weight (kg)	38	73 (63—85)
Height (cm)	38	166 (162—178)
Body Mass Index (kg/m^2^)	38	26.6 (23.6—30.9)
Body Mass Index (kg/m^2^) ≤ 18.5, *n* (%)		4 (10.5)
Body Mass Index (kg/m^2^) > 30.0, *n* (%)		11 (28.9)
Assistance with self-care activities, *n* (%)	39	26 (66.7)
Smoking current (%)	39	6 (15.4)
≥1 fall in preceding year, *n* (%)	39	22 (56.4)
Malnourished, *n* (%)	39	25 (64.1)
Health Related Quality of Life (EQ-5D-5L)	38	0.69 (0.50—0.78)
*Functional measures*	39	
Handgrip strength (kg), Males	39	32 (19.4—37.5)
Handgrip strength (kg), Females	39	17 (14.5—21.9)
*Cognitive measures*		
Short Orientation-Memory-Concentration test	38	20 (18—22)
*Biomarkers*		
eGFR (mL/min/1.73 m^2^)	39	46 (36—65)
<60 mL/min/1.73 m^2^		25 (64.1)
<30 mL/min/1.73 m^2^		5 (12.8)
Fi-OutRef	38	6 (4—7)
*Hospitalization*		
Length of stay (days)	39	3 (1—7)
Discharged from hospital department	39	
Cardiovascular, *n* (%)	—	10 (25.6)
Emergency department, *n* (%)	—	15 (38.5)
Endocrinology, *n* (%)	—	3 (7.7)
Infectious diseases, *n* (%)	—	2 (5.1)
Respiratory, *n* (%)	—	9 (23.1)
*Number of medications*	39	
Long-term medications	39	10 (8—12)
“As needed” medications	39	2 (1—3)
5–9 in total, *n* (%)	—	18 (18.0)
≥10 total, *n* (%)	—	32 (82.0)
STOPP criteria medication		5 (3—5)
Renal risk medication, *n* (%)		4 (3—5)

CRP, C-reactive protein; eGFR, estimated glomerular filtration rate; IQR, interquartile range; STOPP, Screening Tool of Older Persons’ potentially inappropriate Prescriptions.

**Table 2 jcm-09-00348-t002:** Changes in Medication Appropriateness (MAI) and Assessment of Underutilization Index (AOU) score between admission and 30 days after discharge (*n =* 39).

	Number (%) of Patients
*MAI score*	
≥10 MAI point improvement	16 (41)
≥5 MAI point improvement	25 (64)
≥3 MAI point improvement	28 (72)
≥1 MAI point improvement	34 (87)
The MAI score stayed the same	1 (3)
The MAI score deterioration	4 (10)
*AOU score*	
AOU score improvement	11 (28)
AOU score stayed the same	27 (69)
AOU score deterioration	1 (3)

**Table 3 jcm-09-00348-t003:** Individual Medication Appropriateness Index (MAI) criteria deemed inappropriate at admission, after intervention, and 30 days after discharge.

	Admission	After Intervention	30 Day Follow-Up
Total number of medications	454	430	444
MAI criteria	*n* (%)	*n* (%)	*n* (%)
Not indicated	48 (10.7)	26 (6.0)	31 (6.9)
Not effective	41 (9.0)	31 (7.2)	33 (7.4)
Dose incorrect	62 (13.7)	32 (7.4)	33 (8.1)
Direction incorrect	8 (1.8)	2 (0.5)	2 (0.5)
Direction impractical	9 (2.0)	1 (0.2)	2 (0.5)
Drug–drug interaction	12 (2.6)	3 (0.7)	1 (0.2)
Drug–disease interaction	23 (5.1)	5 (1.2)	4 (0.9)
Drug duplication	5 (1.1)	3 (0.7)	2 (0.5)
Incorrect duration	63 (13.9)	36 (8,4)	42 (9.3)
Cost	53 (11.7)	29 (6.7)	33 (7.4)

**Table 4 jcm-09-00348-t004:** Most frequently deprescribed medications.

Drug Class	Frequency *n* (%)
Proton pump inhibitor	15 (22.7)
Sedatives *	11 (16.7)
Antihypertensives	10 (15.1)
Opioids	5 (7.6)
Acetaminophen	5 (7.6)
NSAID	4 (6.1)
Statins	4 (6.1)
Anticoagulants	3 (4.5)
Other	9 (13.6)
In total	66

* Sedatives includes: nonbenzodiazepines, benzodiazepines and first-generation antihistamines. NSAID, nonsteroidal anti-inflammatory drug.

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
