# Peer review of "A Collaborative Medication Review Including Deprescribing for Older Patients in an Emergency Department: A Longitudinal Feasibility Study"

_jcm, 2020, doi:10.3390/jcm9020348_

Round 1
Reviewer 1 Report
This is a longitudinal feasibility study of the effect of collaborative medication review and deprescribing (clinical pharmacist and geriatrician team) on post-discharge and at 30 day medication-related measures for elderly polypharmacy patients presenting to the emergency department of one Danish hospital.
The manuscript is very well written and the study’s methodology is well described. The abstract presents an excellent synopsis of findings and the key words are well chosen for the study’s context. Statistical tests are appropriate. References are very relevant and supportive, and limitations are well noted. Citations are in MDPI style.
This is a very novel examination of the potential for collaboration within an ED context. I commend the authors for an excellent contribution to the literature.
Author Response
Response to Reviewer 1 Comments
Thank you for your positive response to our manuscript (Journal of Clinical Medicine- 693461)” A Collaborative Medication Review Including Deprescribing for Older Patients in an Emergency Department: A Longitudinal Feasibility Study”.
We appreciate the positive feedback and your interest in the area.

Reviewer 2 Report
The authors assessed the impact of collaborative medication review focused on deprescribing in older adults in the emergency department. I find this a great undertaking and the study appears to follow an acceptable, well-thought methodology. Below are my comments/suggestions for improvement:
The legend for figure 2 needs to be expanded a bit more for clarity for wider audiences. Please describe the figure a bit more. In table 4, no entries for nonbenzodiazepines and first-generation antihistamines.Author Response
Response to Reviewer 2 Comments
Thank you for your positive response to our manuscript (Journal of Clinical Medicine- 693461)” A Collaborative Medication Review Including Deprescribing for Older Patients in an Emergency Department: A Longitudinal Feasibility Study”.
We appreciate the constructive feedback and have incorporated the comments and suggestions into our revised manuscript. We hope that you will find the changes and our answers to the comments satisfactory and will consider the revised manuscript for publication.
Point 1: The legend for figure 2 needs to be expanded a bit more for clarity for wider audiences. Please describe the figure a bit more.
Response 1:
We agree with your observations and have elaborated the description of Figure 2 a bit more.
“The appropriateness of prescribed medication was assessed by the total MAI score at each time point for each patient and are shown in Figure 2” Line 290-291.
AND
“Overall, the results from Figure 2 show that the intervention increases the prescribing appropriateness and the changes were sustained 30 days after discharge.” Line 295-297.
Finally, we have enlarged the text size in Figure 2.
Point 2: In table 4, no entries for nonbenzodiazepines and first-generation antihistamines.
Response 2:
Thank you for this essential observation.
We agree with your observations and have now collected nonbenzodiazepines, benzodiazepines and first-generation antihistamines in one group called Sedatives. Table 4 has been updated. Line 348-349.